# Attend to the Active: Structure-Aware Dynamic Attention in LLMs for Compositional Instruction Following

**Fangrui Lv**[1]  **Yulei Qin**[2]  **Ruixin Hong**[1]  **Jian Liang**[3]  **Jinyang Wu**[1]
**Ke Li**[2]  **Xing Sun**[2]  **Changshui Zhang**[1,*]
[1] Beijing National Research Center for Information Science and Technology (BNRist)
Department of Automation, Tsinghua University, Beijing, P.R.China
[2] Tencent Youtu Lab    [3] Kuaishou Technology
lvfr23@mails.tsinghua.edu.cn  {yuleiqin,tristanli,winfredsun}@tencent.com
hongrx16@tsinghua.org.cn  liangjian03@kuaishou.com
wu-jy23@mails.tsinghua.edu.cn  zcs@mail.tsinghua.edu.cn

## Abstract

Large language models (LLMs) have demonstrated strong instruction-following capabilities; however, they often struggle with compositional instructions that involve multiple interleaved yet logically independent sub-tasks. These sub-tasks are typically organized in mutually exclusive structures, such as branching, chaining, or paralleling, where only one sub-task should be active at each generation step, while the others remain dormant. Despite their inactivity, dormant sub-tasks can inadvertently attract the model's attention due to structural entanglement within the input context or intermediate representations, leading to interference that compromises output fidelity. To address this challenge, we propose ATA, a structure-aware dynamic attention mechanism grounded in compositional structures, which dynamically identifies the active sub-task during generation while suppressing attention to inactive ones. By precisely steering the model's focus, ATA mitigates interference and explicitly enhances model adherence to the active sub-task. Importantly, ATA operates within a single forward pass without requiring parameter updates. Extensive experiments show that ATA consistently enhances LLMs' instruction-following ability across various compositional structures, effectively mitigating attention distraction and demonstrating a strong generalization ability.

## 1 Introduction

Large language models (LLMs) have remarkably succeeded in numerous natural language processing tasks (Zhao et al., 2023; OpenAI, 2023). With their growing capabilities, LLMs are increasingly applied to handle compositional instructions involving multiple sub-tasks in real-world scenarios (Ouyang et al., 2022; Xu et al., 2024). However, existing research has predominantly focused on compositional instructions requiring the simultaneous completion of interleaved sub-tasks (He et al., 2024b; Sun et al., 2024), the exploration of complex composition structures remains limited. Consider the example in Figure 1 (left), the compositional instruction comprises two independent sub-tasks, but only one should be executed according to the condition judgement. The model should accurately identify and faithfully follow the selected active branch while avoiding interference from the unselected inactive one. Existing LLMs, however, often exhibit degraded performance in such intricate structures due to spurious attention to inactive sub-tasks (Wen et al., 2024). We attribute this limitation to the model's tendency to diffusely attend across the entire input context (Zhang et al., 2024b; Zhu et al., 2024), making it susceptible to distraction from irrelevant sub-tasks. This inability to selectively focus often leads to ambiguous reasoning or inconsistent outputs, ultimately undermining the model's ability to follow complex compositional instructions (Wen et al., 2024).

Building on prior work in the community (Hayati et al., 2025; Wen et al., 2024), we identify three prototypical composition structures commonly observed in compositional instructions, as illustrated in Figure 2: i) **Chaining**-sequential execution of sub-tasks, ii) **Branching**-conditional selection of

---

*Corresponding author.

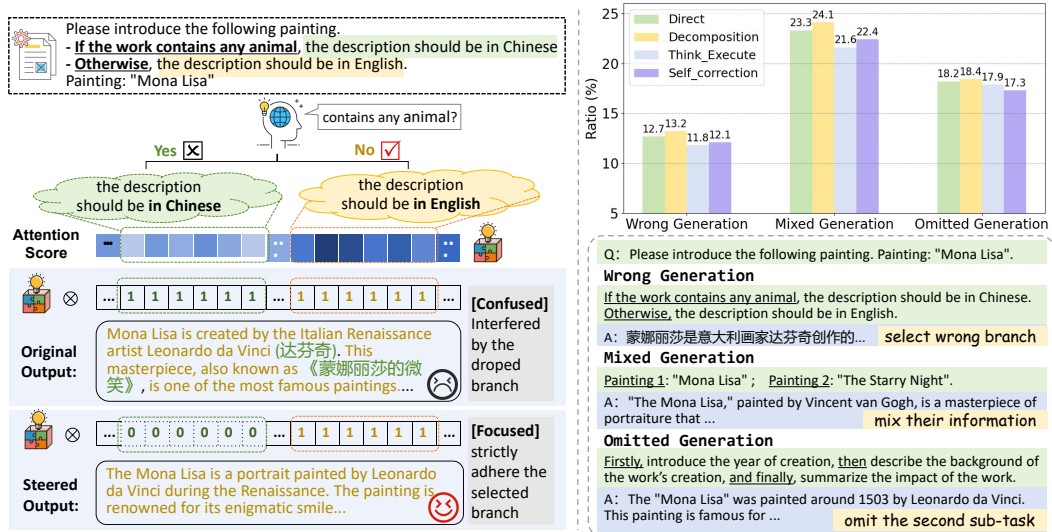

Figure 1: (left) LLMs with original attention distributed across the entire input or with steered attention eliminating the interference from inactive sub-tasks. (right) Distribution and examples of the error types in compositional instructions generated by Llama3-8b.

sub-tasks, and iii) **Paralleling**-parallel stacking of independent sub-tasks. Our systematic analysis reveals that these sub-tasks, although interleaved, are often logically mutually exclusive. That is, only one sub-task (e.g., the selected branch, the currently executed chain or parallel element) should be active and govern the output at each generation step, while others remain dormant and may cause interference. It is worth noting that the status of sub-tasks (active vs. inactive) can dynamically shift throughout the generation. While existing approaches have made progress through techniques such as fine-tuning (Jiang et al., 2024; He et al., 2024a), high-level planning (Chae et al., 2024; Wang et al., 2023) or iterative self-reflection (Sun et al., 2024; Shinn et al., 2023), they often incur substantial computational overhead and still fail to fully suppress attention to dormant sub-tasks during inference. To better understand these limitations, we analyze typical error types in compositional instruction following (Figure 1 (right)): i) **Wrong Generation**: erroneously executing inactive sub-tasks; ii) **Mixed Generation**: conflating multiple exclusive sub-tasks, leading to incoherent or blended reasoning; iii) **Omitted Generation**: skipping one or more required active sub-tasks. These issues largely stem from blended comprehension, where the model conflates mutually exclusive sub-tasks, and from unfocused tracking of dynamic active sub-tasks during generation. Motivated by the observation, we highlight a critical need for handling compositional instructions: **mitigating interference from inactive sub-tasks during both task comprehension and generation.**

Building upon this insight, we propose ATA (**A**ttend **T**o the **A**ctive), a structure-aware dynamic attention framework to tackle the challenges posed by compositional instructions. Recognizing that sub-tasks in such instructions are typically organized in mutually exclusive structures, ATA first analyzes the compositional structure type and extracts constituent sub-tasks. To prevent blending comprehension of multiple sub-tasks during encoding, ATA applies mutual attention masking between exclusive sub-task pairs, ensuring their representations remain disentangled and semantically independent. During generation, ATA dynamically identifies the currently active sub-task for next token prediction by analyzing attention scores, while treating the exclusive ones as dormant. It then selectively suppresses attention to these inactive sub-tasks, minimizing interference and guiding the model's focus toward the active sub-task. By enforcing this structure-aware dynamic attention modulation throughout the generation process, ATA enables LLMs to consistently and pertinently attend to active sub-tasks, thereby reducing ambiguity and improving output fidelity. Notably, ATA operates in a plug-and-play manner with a single forward pass and requires no parameter updates, offering an efficient and versatile technique to boost various LLMs.

It is worth noting that the structure information serves as a reliable guide to conduct safe attention masking in our ATA framework. The attention steering mechanism is confined to the structurally mutually exclusive sub-tasks, ensuring the surrounding context remains untouched. Besides, we propose to perform attention masking at the granularity of independent and complete sub-tasks, such

that the mix across multiple sub-tasks is prevented for disambiguation during masking. In addition, ATA selectively interposes at a subset of the most relevant attention heads, minimizing the risk of global information attenuation. Consequently, ATA ensures the effective elimination of interference from irrelevant information while preserving critical details for the integrity of the overall structure.

To the best of our knowledge, this work is the first to systematically identify the reasons behind the performance degradation of LLMs in tackling compositional instructions, and it also pioneers the introduction of parallel structure in this field. We highlight the role of recalibrated attention in mitigating interference from inactive sub-tasks. Comprehensive experiments across diverse benchmarks and models consistently demonstrate that ATA significantly outperforms various baselines. It effectively shields distraction attacks under various composition structures, showcasing its robustness and versatility. Furthermore, its parameter-free modifications and plug-and-play compatibility with various off-the-shelf LLMs guarantee both efficiency and practicability in real-world scenarios.

## 2  RELATED WORK

**Compositional Instruction Following.** With the increasing complexity in real-world tasks, the study of compositional instructions that involve multiple sub-tasks has garnered significant attention. Prior studies primarily focus on integrating diverse coupled sub-tasks such as semantic (Zheng et al., 2023; Jiang et al., 2024) and format (Xu et al., 2024; He et al., 2024b) constraints, which must be satisfied simultaneously. However, the exploration of compositional instructions comprising multiple independent or mutually exclusive sub-tasks remains limited, where the sub-tasks may be organized with complex structures. Recent efforts, such as Chain-of-Instruction (Hayati et al., 2025), model sequential execution of sub-tasks, while Complexbench (Wen et al., 2024) introduces conditional selected branches. Nonetheless, we are the first to introduce parallel structure, a fundamental yet underexplored paradigm in real-world scenarios. Moreover, existing research lacks systematic investigation into the inherent commonalities across different compositional structures to enhance LLMs' reasoning fidelity. Existing works largely rely on fine-tuning with curated datasets (Xu et al., 2024; Hayati et al., 2025; Jiménez-Navarro et al., 2024), which is resource-intensive. Alternative approaches include iteratively output refinement via self-reflection (Sun et al., 2024; Shinn et al., 2023), decomposition of compositional tasks into sequentially solvable sub-tasks (Wen et al., 2024), and high-level planning followed by step-by-step execution (Chae et al., 2024; Wang et al., 2023). These methods, however, typically depend heavily on high-quality intermediate steps and require multiple inference iterations that lead to inefficiencies. In contrast, our approach innovatively modifies the attention mechanism to handle compositional instructions within a single forward pass. Furthermore, it generalizes effectively across diverse complex structure types and supports plug-and-play functionality, demonstrating superior versatility and generalization ability.

**Attention Steering.** Advances in attention mechanisms have significantly boosted model performance across various tasks. For instance, SampleAttention (Zhu et al., 2024) and SASK (Jin et al., 2024) improve the efficiency of long-context LLM inference through adaptive sparse attention, which compresses redundant information. However, these strategies often risk performance drops due to the loss of critical global information. Besides, SAR (Duan et al., 2024) shifts attention to more relevant components at the token level for better uncertainty quantification, while PASTA and autoPASTA (Zhang et al., 2024a;b) scale up attention on predefined or self-selected input segments to highlight key information. Yet, their emphasized text remains fixed during generation, making them unsuitable for compositional instructions where active sub-tasks shift dynamically. Differently, our proposed ATA constrains attention steering to mutually exclusive sub-task spans, preserving global context. And it dynamically adjusts attention by identifying the active sub-task at each generation step, thus maintaining information integrity while effectively eliminating interference.

## 3  FORMULATION

### 3.1  COMPOSITIONAL INSTRUCTIONS

For a comprehensive analysis, we build on the *chain* and *branch* composition structures from existing literature, which encompass most compositional types in real-world LLM instructions (Wen et al.,

| Structure Type | Definition | Example | Illustration |
|---|---|---|---|
| **Chain** | The generation is required to complete multiple sub-tasks sequentially. | Please introduce "Mona Lisa" briefly. **Firstly,** introduce the year of creation, **then** describe the background of the work's creation, **and finally**, summarize the impact of the work. | $T_1$ $T_2$ $T_3$ |
| **Branch** | The generation is required to select different branches according to certain conditions. | Please introduce the following painting. **- If the work contains any animal**, the description should be in English **- Otherwise**, the description should be in Chinese. Painting: "Mona Lisa" | $S_1(cond_1)$ $T_1$ $T_2$ |
| **Parallel** | The generation is required to complete multiple independent tasks in parallel . | Please introduce the following paintings. **- Painting 1**: "Mona Lisa" **- Painting 2**: "The Starry Night" | $T_1$ $T_2$ |

Figure 2: Structure type in Compositional Tasks. The blue nodes are independent or mutually exclusive sub-tasks, while the grey nodes are other content in the context such as selection condition.

2024; Hayati et al., 2025). We further broaden our coverage by introducing the *parallel* structure, a common yet underexplored composition type. Their formal definitions are as follows.

**Chain.** The generation is required to complete multiple sub-tasks sequentially. Formally, *Chain* contains $m$ sub-tasks $T_1, T_2, \cdots, T_m$, which need to be executed in sequence. The generation of $T_{k+1}$ depends on that of $T_k$ ($k = 1, 2, \cdots, m-1$).

**Branch.** The generation is required to select a specific branch according to given conditions, fulfilling the sub-task associated with the chosen branch. Formally, *Branch* contains $m$ distinct branches, each corresponding to a sub-task $T_1, T_2, \cdots, T_m$ with respective expected outputs $Y_1, Y_2, \cdots, Y_m$. We denote a selection function as $S$ with a range $[1, m]$, taking the selection condition *cond* as input. This function determines which sub-task to execute. Finally, the expected generation is $Y_{S(cond)}$.

**Parallel.** The generation is required to complete multiple sub-tasks independently. Formally, *Parallel* contains $m$ independent sub-tasks $T_1, T_2, \cdots, T_m$, each of which is executed individually with the respective expected outputs $Y_1, Y_2, \cdots, Y_m$. Ultimately, the expected generation is a parallel joint of these individual outputs.

**Complexity.** We define the complexity of a compositional instruction by the number of its structurally mutually exclusive sub-tasks. As this complexity escalates, the challenge of adhering to associated requirements proportionally intensifies. Notably, each sub-task may encompass multiple constraints that must be satisfied concurrently.

In the standard paradigm of LLM generation, a pre-trained LLM is provided alongside a compositional instruction $\mathcal{T}$ comprising $m$ sub-tasks, represented as $\mathcal{T} = \{T_1, T_2, \cdots, T_m\}$. We denote $T_i \perp T_j$ for $i \neq j$ to indicate that they are structurally exclusive. During each generation step, only one sub-task, denoted as $T_{\text{key}}$ (key $\in [1, m]$), is active and should govern the output, while the structurally incompatible sub-tasks $T_i$ that satisfy $T_i \perp T_{\text{key}}$ fall in a dormant and thus irrelevant state. This work aims to dynamically identify and focus on the active sub-task while mitigating interference from inactive ones, enhancing the output fidelity of the complex compositional instructions.

## 3.2 MULTI-HEAD ATTENTION

A typical transformer-based LLM consists of $L$ stacked layers, where each layer contains two sub-modules: multi-head attention (MHA) and a fully connected feed-forward network (FFN). Given the input $X \in \mathbb{R}^{n \times d}$, MHA of the layer $l$ performs the attention function in parallel $H$ heads: $\text{MHA}^{(l)}(X) = \text{Concat}(H^{(l,1)}, ..., H^{(l,H)})W_o$ with

$$H^{(l,h)} = \text{Softmax}\left(A^{(l,h)}\right)V = \text{Softmax}\left(QK^{\top}/\sqrt{d_h}\right)V, \tag{1}$$

where $Q = XW_{q_h}, K = XW_{k_h}, V = XW_{v_h}$, and $W_{q_h}, W_{k_h}, W_{v_h} \in \mathbb{R}^{d \times d_h}$ are learnable projection matrices of head $h$. $d_h$ is typically set to $d/H$. $A^{(l,h)}$ denotes the attention scores at head $h$ in layer $l$.

## 4 METHOD

Our proposed ATA framework seamlessly integrates structure analysis with attention steering. The structure analysis component identifies the composition structure and mutually exclusive sub-tasks. Equipped with the identified structure as guidance, the attention steering component safely identifies the active sub-task and effectively eliminates interference from inactive ones during both task comprehension and generation. For clarity, the overall algorithm is provided in Alg. 1.

### 4.1 STRUCTURE ANALYSIS AND IDENTIFICATION

Given a compositional instruction $\mathcal{T}$, we first need to accurately identify its structure type, i.e., *Chain*, *Branch* or *Parallel*, and extract the corresponding sub-tasks to support attention steering. Although LLMs often struggle with end-to-end execution over complex entangled structures, they exhibit strong capabilities in interpreting and decomposing sub-tasks when guided by well-crafted prompts (Zhang et al., 2024b; Chae et al., 2024). We leverage this by designing prompts that elicit both structural labeling and sub-task enumeration, using the consistent LLM for instruction following:

$$S, T_1, T_2, \cdots, T_m = \mathcal{LLM}(\mathcal{T}|\mathcal{P}), \tag{2}$$

where $S \in [Branch, Chain, Parallel]$ denotes the structure label, $[T_1, T_2, \cdots, T_m]$ represents the identified structurally exclusive sub-tasks within $\mathcal{T}$, and $\mathcal{P}$ is the designed prompt template as below. Additional details and examples are provided in Appendix A.2.

**Remark.** To avoid potential token-level errors in sub-task extraction, we further match each identified sub-task back to its corresponding span in the original input, enforcing faithfulness and preventing error propagation. Empirical evidence shows that LLMs perform well in structure identification across various composition types and complexity (see Appendix B.1). The acquired structure information serves as both guidance and constraint for subsequent safe attention steering, which is confined to sub-task spans to preserve global information integrity. Therefore, minor omissions in sub-task extraction remain non-destructive and only slightly diminish the effectiveness of attention modulation since leaving partial sub-task regions untouched, indicating the robustness of ATA to imperfect structure identification (see Figure 4(b)).

> **Prompt template $\mathcal{P}$ of structure identification**
>
> You are an excellent logic expert. Given a compositional task and the definition of structure types, please determine the structure label of the task and identify the corresponding sub-tasks.
>
> **Structure Type**:
> - Chain: The task is required to complete multiple sub-tasks sequentially.
> - Branch: The task is required to select different branches according to certain conditions.
> - Parallel: The task is required to complete multiple independent sub-tasks in parallel.
>
> Output the structure type of input task (chosen from "Chain", "Branch", "Parallel") and the exactly sub-tasks (without modifying, deleting or adding any original text) in the following json format:
> { "Structure Type": "", "Sub-tasks": [sub-task1, sub-task2, ...] }
>
> Task: {input task}

### 4.2 STRUCTURE-AWARE DYNAMIC ATTENTION

Leveraging the structure information, ATA subsequently mitigates mutual interference between exclusive sub-tasks (Step 1) and suppresses interference of inactive sub-tasks during inference (Steps 2–3), thus preventing task blending and enhancing faithful generation. See Figure 3 for illustration.

**Step 1: Mutually Exclusive Masking.** Given the identified structure type $S$ and the mutually exclusive sub-tasks $[T_1, T_2, \cdots, T_m]$, we can obtain the exclusive sub-task pairings, i.e., $T_i \perp T_j, \forall i \neq j, i, j \in [1, m]$. Considering their structural exclusivity, information interaction between them may introduce blending comprehension during encoding. Specifically, sequential sub-tasks should not consider subsequent ones; cross attention between sub-tasks corresponding to different branches can cause ambiguity; an independent task referencing others may hinder self-understanding. Therefore, we propose to mask attention between the corresponding token sets of each exclusive sub-task pair, thereby mitigating potential mutual interference. Inspired by the attention mechanism in LLMs that restricts each token from attending to subsequent tokens via incorporating a negative infinity

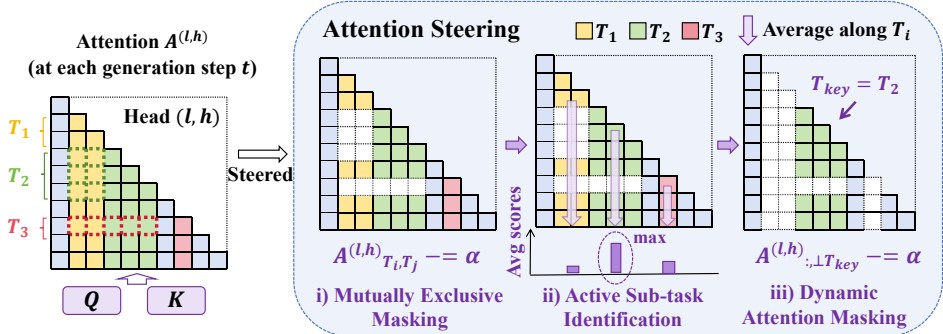

Figure 3: The illustration of structure-aware dynamic attention steering. At each generation step $t$, steered attention replaces the original attention with a three-stage process: 1) attention between mutually exclusive sub-tasks is masked to avoid blending comprehension; 2) computing the average attention score for each sub-task and identifying the highest-scoring one as active; 3) attention to sub-tasks exclusive with the active one is masked to mitigate interference from irrelevant information.

value (OpenAI, 2023), we similarly adopt a bias matrix $M$ to restrict mutual attention accordingly:

$$H^{(l,h)} = \text{Softmax}\left(A^{(l,h)} + M^{(l,h)}\right)V, \tag{3}$$

$$M^{(l,h)}(T_i, T_j) = \begin{cases} -\alpha, & T_i \perp T_j \\ 0, & \text{otherwise} \end{cases}. \tag{4}$$

Here, $\alpha \in (0, \inf)$ is a constant attention bias controlling the masking degree, from none to complete masking. $M^{(l,h)}(T_i, T_j)$ denotes the attention bias matrix $M$ indexed by token sets of $T_i$ and $T_j$. After applying $Softmax(\cdot)$, attention scores between exclusive sub-task pairs are scaled down by $exp(\alpha)$, effectively minimizing interference between mutually exclusive sub-tasks and enabling independent comprehension. As empirically studied in Section 5.3, we set $\alpha = \log 100$ across all experiments.

**Step 2: Active Sub-task Identification.** To suppress interference from inactive sub-tasks during generation, we first identify the active sub-task $T_{key}$ at generation step $t$, i.e., the one that dominates the model's focus and guides next token prediction. Specifically, we score each sub-task $T_i$ by averaging the attention its associated tokens have attracted from subsequent tokens:

$$score(T_i, t) = \frac{1}{|T_i|} \sum_{k \in T_i} \frac{1}{t-k} \sum_{k \leq q \leq t} A^{(l,h)}(q, k), \tag{5}$$

where $|T_i|$ denotes the number of tokens associated with $T_i$ and $k$ is an associated token. $A^{(l,h)}(q, k)$ represents the attention matrix $A^{(l,h)}$ indexed by the query preceding the next predicted token and the token $k$ in $T_i$. The sub-task with the highest score is identified as the active sub-task $T_{key}$. Notably, the score excludes influence from other sub-tasks since our mutually exclusive masking strategy. To enhance the robustness of this identification, we draw upon the entropy formula from information theory (Chen et al., 2019), $H(X) = -\sum_{i=1}^{n} p(x_i) \log p(x_i)$, which measures the uncertainty of a distribution. Accordingly, we impose an additional constraint on the entropy of normalized scores across all sub-tasks when identifying $T_{key}$:

$$T_{key} = argmax_{T_i}(score(T_i, t)), \quad s.t. H([score(T_1, t), score(T_2, t), ...]) < \epsilon, \tag{6}$$

where $\epsilon = \gamma \log(m)$ scales with the number of sub-tasks $m$ to adapt to instruction complexity, $\gamma$ is empirically set to 0.5. Since lower entropy corresponds to higher confidence, this constraint serves as a confidence measure. When the entropy exceeds the threshold, ATA keeps the previous masking state, avoiding unreliable active sub-task switches and thus ensuring deterministic and stable identification.

**Remark.** Considering the instruction's global structure, ATA augments the above score-based active sub-task identification with an active control mechanism to ensure safe progression. For *Chain/Parallel* tasks, it permits only sequential active sub-task transitions, i.e., $T_1 \rightarrow T_2 \rightarrow \cdots$. Skip or revisit is rejected and the previous masking state is preserved, preventing premature or illogical switches. For *Branch* tasks, once a branch is selected, it keeps a consistent active sub-task throughout generation to ensure faithful execution. This mechanism mitigates errors from occasional attention drift, and the ablation study in Table 2 ("w/o Active Control") demonstrates its effectiveness.

**Step 3: Dynamic Attention Masking.** Given the active sub-task $T_{key}$ at each generation step $t$, other structurally exclusive sub-tasks are considered as inactive. To eliminate their interference, ATA masks their attention scores with a constant attention bias, building upon the matrix $M^{(l,h)}$ in Eq 4:

$$H^{(l,h)} = \text{Softmax}\left(A^{(l,h)} + M^{(l,h)}\right)V, \tag{7}$$

$$M^{(l,h)}(:,T_j) = \begin{cases} -\alpha, & T_j \perp T_{key} \\ 0, & \text{otherwise} \end{cases}, \tag{8}$$

where $M^{(l,h)}(:,T_j)$ denotes the attention bias matrix $M$ indexed by token sets of $T_j$. After Softmax($\cdot$), the attention of inactive sub-tasks are suppressed by $\exp(\alpha)$, while those corresponding to $T_{key}$ are indirectly upweighted. The $\alpha$ is set consistently with Eq. 4 as log 100. Consequently, this mechanism effectively steers the model to focus exclusively on the active sub-task while minimizing interference from irrelevant information, ensuring faithful and consistent following of compositional instructions.

**Discussion.** The attention steering mechanism operates during a single forward pass and requires no parameter modification. Though it incurs a minor runtime overhead due to the attention edition, we emphasize that this overhead is minimal (less than 7% as shown in Appendix B.3). Importantly, the attention modification is confined to the span of mutually exclusive sub-tasks, ensuring the surrounding context remains intact and preventing global information loss (see Section 5.3). Moreover, attention modules exhibit diverse token-attending patterns across different heads (Voita et al., 2019; Michel et al., 2019; Zhang et al., 2024a). Involving all attention heads in steering could risk degrading global context understanding and lead to model collapse, as demonstrated experimentally in Section 5.3. Thus, we heuristically select a small subset of attention heads that are particularly responsive to active sub-tasks to conduct attention steering. Details can be found in Appendix A.3.

## 5 EXPERIMENT

### 5.1 SETUPS

**Benchmark.** We evaluate ATA across three composition structure types: (i) **Chain-based** benchmark, comprising 325 instructions with sequential sub-tasks of length 2 to 3; (ii) **Branch-based** benchmark, consisting of 435 instructions that require selecting one branch to follow according to conditions varying from single-layer to nested multi-layers. Both chain-based and branch-based benchmarks are subsets of Complexbench (Wen et al., 2024); (iii) **Parallel-based** benchmark, including 450 instructions formed by concatenating multiple independent tasks from gsm8k (Cobbe et al., 2021).

**Baseline.** We compare ATA with various baselines commonly adopted for handling compositional instructions, including general prompting, decomposition-based, multi-turn dialogue, and planning-based techniques. Direct I/O represents the standard method of interacting with LLMs, where users input a single instruction. CoT Prompting (Wei et al., 2022) encourages a chain-of-thought reasoning process to improve understanding. Decomposition (Wen et al., 2024) breaks down complex instructions into sequential sub-tasks to achieve the final result. Self-correction (Sun et al., 2024) first generates initial outputs and then iteratively refines errors based on its own feedback. Think-execute (Chae et al., 2024) requires LLMs to first outline a plan and subsequently execute it.

**Experiment Setups.** We implement ATA on two pre-trained LLMs: LLaMA3-8B (Touvron et al., 2023) and Mistral-7B (Jiang et al., 2023). Hyper-parameter sensitivity analysis indicates that ATA is robust to mask degree $\alpha$ (see Section 5.3), which we therefore fix it at log 100 for all experiments. For attention head selection, we choose 50 heads for branch-based benchmark and 20 heads for chain and parallel-based benchmarks, with selection criteria detailed in Appendix A.3. Evaluation for chain and branch tasks follows Complexbench standards while precision metric is adopted for parallel tasks.

### 5.2 MAIN RESULTS

The results in Table 1 demonstrate that ATA effectively steers models to attend to the active sub-task while eliminating interference from irrelevant information, ensuring better output fidelity with the compositional instructions. Specifically, while Llama3-8b achieves only 59.80% compliance in the parallel structure, and baseline methods perform marginally exceeded or even worse, ATA yields a remarkable 10.11% improvement by steering attention appropriately. Moreover, ATA achieves 6.45% accuracy when applied to Mistral-7B in the parallel structure, surpassing all the baselines even though they explicitly incorporate iterative feedback. The failure of baselines may be attributed to

Table 1: Performance of ATA and baselines on compositional instructions. The second best performance is underlined, while the best performance is **bold**.

| Structure Type | Chain | | | Branch | | | | Parallel | | | | All |
|---|---|---|---|---|---|---|---|---|---|---|---|---|
| Complexity | 2 | 3 | Avg. | 2 | 3 | ≥ 4 | Avg. | 2 | 3 | ≥ 4 | Avg. | Avg. |
| *Llama-3-8B-Instruct* | | | | | | | | | | | | |
| Direct I/O | 66.93 | 56.82 | 59.21 | 59.74 | 55.26 | 48.32 | 54.63 | 61.63 | 60.17 | 55.83 | 59.80 | 57.88 |
| CoT Prompting | 64.05 | 55.73 | 57.69 | 56.62 | 55.22 | 53.39 | 55.02 | 64.17 | 67.05 | 62.08 | 64.97 | 59.23 |
| Decomposition | 61.19 | 52.61 | 54.63 | 53.88 | 51.49 | 45.66 | 50.04 | 62.98 | 66.74 | 56.26 | 62.58 | 55.75 |
| Think-Execute | 65.94 | 59.28 | 60.85 | 57.49 | 52.26 | 50.25 | 52.92 | 63.4 | 68.33 | 57.09 | 63.14 | 58.97 |
| Self-correction | 66.31 | 60.33 | 61.74 | 61.27 | 56.38 | 50.21 | 55.24 | 63.37 | 67.92 | 51.39 | 61.25 | 59.41 |
| **ATA** | **69.26** | **61.04** | **62.98** | **64.38** | **58.42** | **53.79** | **58.74** | **72.50** | **71.03** | **65.42** | **69.91** | **63.88** |
| *Mistral-7B-Instruct* | | | | | | | | | | | | |
| Direct I/O | 57.41 | 55.68 | 56.31 | 55.40 | 49.37 | 41.19 | 48.83 | 37.87 | 35.24 | 31.50 | 34.89 | 46.67 |
| CoT Prompting | 55.78 | 53.64 | 54.19 | 57.14 | 51.24 | 42.98 | 50.45 | 40.45 | 39.07 | 33.89 | 37.62 | 47.42 |
| Decomposition | 52.50 | 51.17 | 51.62 | 48.74 | 44.26 | 37.59 | 43.77 | 40.26 | 38.44 | 32.15 | 37.41 | 44.27 |
| Think-Execute | 56.48 | 55.32 | 55.89 | 57.33 | 50.15 | 41.83 | 49.26 | 39.86 | 37.95 | 32.61 | 36.72 | 47.28 |
| Self-correction | 59.62 | 55.94 | 56.94 | 56.26 | 47.19 | 40.64 | 47.65 | 38.62 | 36.43 | 31.88 | 35.22 | 46.60 |
| **ATA** | **61.73** | **57.28** | **58.37** | **60.44** | **52.93** | **43.76** | **52.16** | **44.68** | **41.77** | **37.59** | **41.34** | **50.62** |

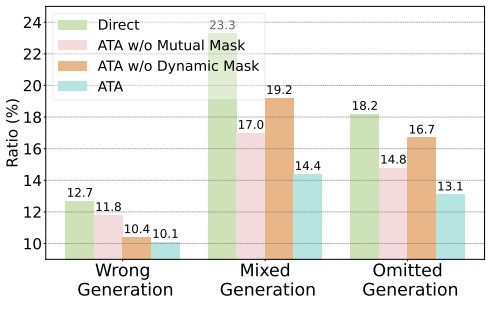
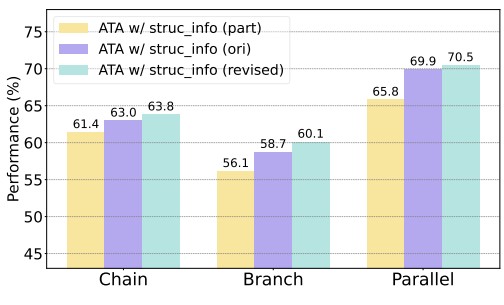

(a) Reduction of Generation Error (↓).  (b) Robustness of ATA to Structure Identification.

Figure 4: (a) The generation error reduction with different ATA variants. (b) Robustness of ATA to different quality of structure identification. Both results are produced by Llama3-8b model.

their reliance on high-quality feedback and their tendency to distribute attention across all mutually exclusive sub-tasks, potentially introducing interference. The consistent improvement of ATA across all benchmarks and models underscores its validity and superiority. Additionally, Appendix C presents a few examples generated by Llama-3-8B using ATA, further demonstrating its efficacy.

## 5.3 ANALYSIS EXPERIMENTS

**Ablation Study.** To investigate the effectiveness of each component within ATA, we conduct ablation studies with results shown in Table 2. i) Importance of structure information: The variants of ATA without structure information select active token spans across the entire text, leading to a notable performance drop. This is likely due to masking essential global context or covering mixed and incomplete sub-tasks, highlighting the necessity of structural guidance for safe and effective attention steering, which restricts steering spans and granularity. ii) Importance of attention masking modules: ATA outperforms variants using only mutually exclusive or dynamic masking, and achieves superior error reduction across all generation error types in Figure 4(a), validating their efficacy. Specifically, the mutual mask effectively prevents blending comprehension between mutually exclusive sub-tasks, thus eliminating mixed execution and wrong decisions. The dynamic mask eliminates interference from inactive sub-tasks during generation, ensuring faithful execution of active sub-tasks. iii) Importance of valid active sub-task identification: We have employed an active control strategy in ATA to ensure valid active sub-task identification. This strategy enforces consistent active sub-tasks

Table 2: Ablation Study of ATA.

| METHOD | CHAIN | BRANCH | PARALLEL |
|---|---|---|---|
| DIRECT | 59.21 | 54.63 | 57.88 |
| **ATA** | **62.98** | **58.74** | **69.91** |
| - w/o STRUCTURE INFO | 60.45 | 55.28 | 60.35 |
| - w/o MUTUAL MASK | 61.82 | 57.03 | 67.42 |
| - w/o DYNAMIC MASK | 60.74 | 56.65 | 64.31 |
| - w/o ACTIVE CONTROL | 61.14 | 57.23 | 66.82 |

Table 3: Effectiveness of Attention Steering.

| STEERING (ST.) STRATEGY | CHAIN | BRANCH | PARALLEL |
|---|---|---|---|
| DIRECT (NO ST.) | 59.21 | 54.63 | 57.88 |
| SAMPLEATTENTION | 60.47 | 51.92 | 57.71 |
| PASTA | 61.24 | 54.16 | 60.43 |
| **ATA** | **62.98** | **58.74** | **69.91** |
| - w/ MISGUIDED ST. | 56.43 | 47.86 | 54.12 |
| - w/ RANDOM ST. | 57.38 | 51.04 | 56.81 |

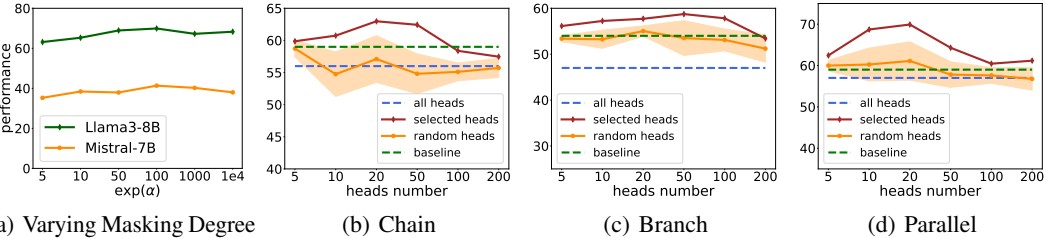

| (a) Varying Masking Degree | (b) Chain | (c) Branch | (d) Parallel |
|---|---|---|---|

Figure 5: (a) the sensitivity of ATA to the masking degree hyperparameter $\alpha$; (b)(c)(d): The performance of ATA when applied to varying numbers of attention heads.

during generation for branching tasks and sequential progression for chaining or paralleling tasks. The performance degradation without it indicates its role in reducing occasional identification errors.

**Robustness of ATA to Structure Identification.** In Figure 4(b), we evaluate ATA's robustness to structure information quality under three settings: 1) human-revised (ground-truth) information to serve as a performance upper bound, 2) original information from the raw output of our module, and 3) partial (intentionally degraded) information to assess robustness to imperfect identification. ATA consistently achieves performance gains even with imperfect structure information, with only a slight reduction in effectiveness. This non-destructive character arises because attention steering, despite its imperfections, enables models to focus on a smaller, relevant scope while preserving contextual information in unextracted sub-tasks. Although mild intra-group confusion may occur, it is preferable to the global entanglement seen with the original instructions, demonstrating our robustness.

**Validity of Attention Steering.** To highlight the unique strength of our attention steering mechanism in handling compositional instructions, we compare ATA with other attention modulation methods, PASTA (Zhang et al., 2024a) and SampleAttention (Zhu et al., 2024), as shown in Table 3. Results show that these methods are ill-suited for dynamic active sub-task transitions. SampleAttention suffers performance drops due to disrupted information from its global sparse attention. Though PASTA (adapted to emphasize special sub-task tags (e.g., 'if', 'first', 'problem 1', etc)) yields slight improvements, it still underperforms compared to ATA because its static emphasis fails to capture evolving reasoning demands. To provide more direct evidence of validity, we further measure the consistency between ATA's identified active sub-tasks and the ground-truth, achieving 81.2%, 75.1%, and 86.4% for Chain, Branch, and Parallel structures, respectively. Besides, steering the model toward misguided (inactive) or random sub-tasks results in significant performance degradation, strongly confirming that ATA's precise focus alignment with the active sub-tasks drives its improvements.

**Generalization on Nested Structures & Larger LLMs.** Table 4 evaluates ATA on complex composi-

-tional instructions with nested structures involving multiple types. The consistent improvements demonstrate the effectiveness of precise attention steering in maintaining focus on active sub-tasks and reducing inference ambiguity. While the gains are relatively modest compared to single-structure instructions, this capability is crucial for achieving better adherence in such complex scenarios. Additionally, we evaluate ATA on a larger-scale LLM, Llama3-13B, where notable improvements further validate its generalization and versatility.

Table 4: Generalization on Nested Composition Structures and Larger LLMs.

| | STRUCTURE TYPE | DIRECT | ATA |
|---|---|---|---|
| LLAMA3-8B | CHAIN & BRANCH | 48.42 | **51.95** |
| | CHAIN & PARALLEL | 50.17 | **54.72** |
| | BRANCH & PARALLEL | 46.83 | **48.41** |
| LLAMA3-13B | CHAIN & BRANCH | 52.26 | **56.64** |
| | CHAIN & PARALLEL | 56.48 | **59.30** |
| | BRANCH & PARALLEL | 51.65 | **53.88** |

**Attention Head Profiling.** To select appropriate attention heads for steering, we employ a simple yet effective heuristic strategy. Specifically, we analyze a set of high-quality question–response pairs to identify heads that consistently focus exclusively on active sub-tasks across different instructions, as detailed in Appendix A.3. We compare steering the selected heads against (i) all heads and (ii) random heads, as shown in Figures 5(b) to 5(d). Steering all heads degrades performance due to excessive masking, while steering random heads leads to high performance variance as attention heads serve diverse functions (Clark et al., 2019; Voita et al., 2019), demonstrating the necessity of this strategy. Additionally, sensitivity analysis on varying numbers of selected heads reveals a trade-off between focusing on active sub-tasks and maintaining overall structural information. Empirically, we apply ATA to a moderate number of heads from 20 to 50, balancing faithfulness and generation quality.

**Sensitivity To Masking Degree.** We analyze ATA's sensitivity with varying masking degree $\alpha$ from $log5$ to $log10000$. Figure 5(a) shows that ATA exhibits robust across this range, as minimal residual attention to inactive sub-tasks effectively suppresses interference. This insensitivity to specific value allows us to fix $\alpha$ at $log100$ in all experiments, balancing simplicity and effectiveness.

**Preservation of General Response Capability.** To assess whether ATA compromises the model's fundamental language ability, we conduct a GPT-4o-based evaluation following Stolfo et al. (2025). For each instruction, we extract a base query and generate corresponding scoring questions to measure general response quality. Table 5 shows the average differences in quality score relative to a

Table 5: Average Differences in Response Quality Scores.

|  | CHAIN | BRANCH | PARALLEL |
|---|---|---|---|
| DIRECT I/O (BASE QUERY) | 0 | 0 | 0 |
| DIRECT I/O (w/ INSTRUCTION) | -0.17 | -0.41 | +0.06 |
| ATA (w/ INSTRUCTION) | **-0.21** | **-0.28** | **+0.09** |

simple Baseline (Base Query), isolating the impact of adding complex instructions and ATA's attention masking. While complex instructions can degrade the general response quality due to constraint compliance, ATA shows a robust performance: it introduces negligible additional degradation on *Chain*, mitigates the quality drop on *Branch* by masking irrelevant branches, and slightly improves *Parallel* by reducing interference between independent sub-tasks, enabling more focused and coherent generation. We attribute this to precise masking confined to sub-task spans and targeted steering over a small set of structure-relevant heads, preserving most heads responsible for language ability.

## 6 CONCLUSION & LIMITATION

This work aims to tackle the challenge faced by LLMs in following compositional instructions. We systematically analyze the essential characteristics inherent in chaining, branching, and paralleling structure types, revealing that only one sub-task should be active at each generation step, and residual attention to inactive sub-tasks can introduce interference. To mitigate this, we propose ATA framework, which dynamically steers the model's attention to focus on active sub-task during generation while masking the inactive ones, thereby eliminating potential interference. Comprehensive experiments across diverse structure types validate that ATA significantly enhances reasoning fidelity to compositional instructions, highlighting the potential of attention-level steering as a lightweight yet effective mechanism. Moreover, ATA is a plug-and-play mechanism that can be integrated with various on-the-shelf LLMs and orthogonal methods to further improve their performance.

**Limitation.** ATA is tailored for complex compositional instructions with explicit sub-task structure. In challenging scenarios where structure is implicit, it needs to be paired with other structure-discovery techniques, with ATA serving as a general-purpose attention-control module to improve attention adherence and reasoning fidelity. We will explore structure-agnostic attention steering in future work to further broaden ATA's applicability and hope it can bring some inspiration to the community.

## ACKNOWLEDGMENTS

This work is supported by the National Science and Technology Major Project (No. 2022ZD0114903) and the Natural Science Foundation of China (NSFC. No. 62476149).

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

# A  IMPLEMENTATION DETAILS

## A.1  ALGORITHM OF ATA

For clarity, the algorithm of proposed ATA is provided in Alg. 1:

---
**Algorithm 1** The algorithm of ATA
---
**Input**  A compositional task $\mathcal{T}$, an LLM $\mathcal{LLM}$, a prompt template $\mathcal{P}$, the selected head set $\mathcal{H}$.
 1:  Generate structure type $S$ and mutually exclusive sub-tasks $[T_1, T_2, \cdots, T_m]$ within $\mathcal{T}$ by Eq. 2;
 2:  **for** *each generation step t* **do**
 3:      Mask the mutual attention between exclusive sub-task pairs by Eq. 4;
 4:      Calculate contribution score $score(T_i, t)$ for each sub-task $T_i$ by Eq. 5;
 5:      Identify the active sub-task $T_{key}$ by Eq. 6;
 6:      Mask the attention to inactive sub-tasks that structurally exclusive with $T_{key}$ by Eq. 7;
 7:  **end for**
**Output:**  The final generation

---

We implement all algorithms using PyTorch (Paszke et al., 2019) and Huggingface (Wolf et al., 2019).

## A.2  DETAILED PROMPT TEMPLATE FOR STRUCTURE ANALYSIS

Table 7 provides the prompt template we used for structure analysis and identification in Section 4. To improve identification quality, we use 3 manually constructed in-context examples in the prompt.

## A.3  ATTENTION HEAD SELECTION STRATEGY

Since various attention heads function diversely, steering different heads yields dramatically different performance. To identify the effective heads, PASTA (Zhang et al., 2024a) proposes a greedy selection strategy that evaluates the steering performance of each head on small validation sets of multiple tasks and selects the heads that yield the best performance. This greedy strategy requires evaluating $L \times H$ times, where $L$ represents the number of attention layers and $H$ represents the number of heads within each layer, resulting in non-trivial overheads especially for large models. AutoPASTA (Zhang et al., 2024b) enhances this strategy by adopting a coarse-to-fine search approach, transitioning from layer-level to head-level selection. Despite its refinement in efficiency, the method still incurs high computational costs with $l \times H + L$ forward passes ($l$ is the number of coarsely selected layers). To address this, we propose a simple yet effective heuristic selection strategy within a single forward pass as an alternative. Specifically, we begin by generating ideal outputs for small validation sets of multiple tasks using GPT-4. These outputs are then concatenated with the corresponding input context and processed by an inference model, such as Llama3-8B, to obtain attention scores for each head. Our objective is to select heads that functioned to be focused exclusively on the active sub-task. Thus, for each data, we measure the attention each head allocates to the mutually exclusive sub-tasks. Heads that exhibit high attention to the active sub-task while showing minimal attention to the inactive ones are shortlisted as candidate heads. Finally, we select the tok-$k$ heads from these candidate, based on their frequency across the entire validation sets. Empirically, we find that a small subset of heads (e.g., 50 out of 1024 heads) is sufficient for ATA to achieve superior performance, demonstrating its efficiency. Table 6 presents the number of heads to be steered by ATA for Llama-8B and Mistral-7B on each benchmark.

Table 6: The number of heads to be steered by ATA.

| Task | Llama3-8B | Mistral-7B |
|---|---|---|
| Chain | 20 | 20 |
| Branch | 50 | 50 |
| Parallel | 20 | 20 |

Table 7: Prompt template for Structure Analysis and Identification.

---

**Prompt template for Structure Analysis and Identification.**

You are an excellent logic expert. Given a compositional instruction and the definition of structure types, please determine the structure label of the task and identify the corresponding sub-tasks.

**Structure Type**:
- Chain: The instruction is required to complete multiple sub-tasks sequentially.
- Branch: The instruction is required to select different branches according to certain conditions.
- Parallel: The instruction is required to complete multiple independent sub-tasks in parallel.

**Output Format**:
Output the structure type of input instruction (chosen from "Chain", "Branch", "Parallel") and the exactly sub-tasks (without modifying, deleting or adding any original text) in the following json format:
{
"Structure Type": "",
"Sub-tasks": [sub-task1, sub-task2, ...]
}

## **Examples**:
**[Instruction 1]** You are playing the role of Su Shi, a Chinese literati from the Northern Song Dynasty.
- When my words are your poem, please reply with the next line. Then write a paragraph of your thoughts on this line of poetry, with coherent emotions.
- When my words are not your poem, please reply with a poem of yours on the same theme.
Note:
1. Always remember that you are playing Su Shi.
2. Your response style should reflect Su Shi's criticism of reality, lamentation of life as a dream, and free-spirited lyrical expression.
3. Your responses should be positive and uplifting, with a sense of heroic spirit.
"""
Me: When will the moon be bright?
Su Shi:
"""
**[Output 1]**
{
"Structure Type": "Branch",
"Sub-tasks": ["please reply with the next line. Then write a paragraph of your thoughts on this line of poetry, with coherent emotions.", "please reply with a poem of yours on the same theme."]
}

**[Instruction 2]** Select one of the following works that represents the "Fauvism" art movement and generate a Xiaohongshu-style article of about 300 words. Then, conduct an art analysis on the other work, with clear and distinct points, and ensure to include the keywords "Cubism" or "Geometricism".
""" Works: "The Style of Cambrai" and "The Dance""""
**[Output 2]**
{
"Structure Type": "Chain",
"Sub-tasks": ["Select one of the following works that represents the "Fauvism" art movement and generate a Xiaohongshu-style article of about 300 words.", "Then, conduct an art analysis on the other work, with clear and distinct points, and ensure to include the keywords "Cubism" or "Geometricism"."]
}

**[Instruction 3]** Solve the following math problems and output in the format:
"Output 1":
"Output 2":
"Output 3":
1. James decides to run 3 sprints 3 times a week. He runs 60 meters each sprint. How many total meters does he run a week?
2. Every day, Wendi feeds each of her chickens three cups of mixed chicken feed, containing seeds, mealworms and vegetables to help keep them healthy. She gives the chickens their feed in three separate meals. In the morning, she gives her flock of chickens 15 cups of feed. In the afternoon, she gives her chickens another 25 cups of feed. How many cups of feed does she need to give her chickens in the final meal of the day if the size of Wendi's flock is 20 chickens?
3. Kylar went to the store to buy glasses for his new apartment. One glass costs $5, but every second glass costs only 60% of the price. Kylar wants to buy 16 glasses. How much does he need to pay for them?"
**[Output 3]**
{
"Structure Type": "Parallel",
"Sub-tasks": ["James decides to run 3 sprints 3 times a week. He runs 60 meters each sprint. How many total meters does he run a week?", "Every day, Wendi feeds each of her chickens three cups of mixed chicken feed, containing seeds, mealworms and vegetables to help keep them healthy. She gives the chickens their feed in three separate meals. In the morning, she gives her flock of chickens 15 cups of feed. In the afternoon, she gives her chickens another 25 cups of feed. How many cups of feed does she need to give her chickens in the final meal of the day if the size of Wendi's flock is 20 chickens?", "Kylar went to the store to buy glasses for his new apartment. One glass costs $5, but every second glass costs only 60% of the price. Kylar wants to buy 16 glasses. How much does he need to pay for them?"]
}

**Input Instruction**: {input instruction}

---

# B  ADDITIONAL ANALYSIS

## B.1  PERFORMANCE OF STRUCTURE IDENTIFICATION.

To evaluate the reliability of LLMs in identifying structure types and extracting sub-tasks, we present the structure identification performance of LLaMA3-8B across compositional instructions of varying complexity in Table 8. Specifically, *Structure-type Precision* measures the accuracy of LLMs in predicting structure types; *Sub-task Recall* indicates the proportion of ground-truth sub-tasks that can be independently identified; and *Sub-task Precision* assesses whether the extracted sub-tasks are semantically complete without unrelated global context. The results demonstrate that LLMs exhibit strong capabilities in structure-type classification and sub-task extraction, maintaining high reliability even for complex compositional instructions. The primary challenge, however, lies in sub-task recall, as multiple sub-tasks may sometimes be grouped. Despite this, the sub-task splits remain contextually precise, that is, the extracted sub-tasks are semantically complete without extraneous content, thereby preventing blend comprehension and ensuring the safety of subsequent attention steering.

Table 8: Performance of Structure Identification.

| COMPLEXITY | 2 | 3 | ≥ 4 |
| --- | --- | --- | --- |
| STRUCTURE-TYPE PRECISION | 100% | 100% | 100% |
| SUB-TASK RECALL | 98.3% | 87.6% | 81.4% |
| SUB-TASK PRECISION | 99.7% | 96.2% | 92.3% |

## B.2  STRATEGIES FOR ACTIVE SUB-TASK IDENTIFICATION.

Identifying the active sub-task at each generation step is pivotal for effective dynamic attention steering. To ensure the confidence of identification, we employ entropy as the evaluation metric in Eq. 6. For comparative analysis, we compare it with Max probability (the highest probability score within sub-instructions) and the Margin confidence (the difference between the highest and second-highest score) as confidence metrics. As demonstrated in Table 9, entropy outperforms the other metrics, likely due to its consideration of the entire distribution, thus offering a more comprehensive assessment of uncertainty.

Table 9: Strategies for Identifying Active Sub-task.

| STRATEGY | CHAIN | BRANCH | PARALLEL |
| --- | --- | --- | --- |
| $\epsilon-$MAX PROBABILITY | 61.74 | 57.91 | 68.62 |
| $\epsilon-$MARGIN CONFIDENCE | 61.45 | 58.16 | 69.27 |
| $\epsilon-$ENTROPY (ATA) | **62.98** | **58.74** | **69.91** |

## B.3  TIME COST.

Considering that ATA modifies attention during model inference, we provide a computational cost analysis in Table 10, presenting the generation time for parallel instructions of varying complexity on Llama3-8b. Our results demonstrate that ATA incurs a minimal time overhead compared to generation, with an increase of less than 7%. Notably, in contrast to approaches requiring finetuning or multiple LLM inferences, ATA performs generation in a single forward pass, highlighting its efficiency.

# C  EXAMPLES OF ATA GENERATIONS

Table 11, 12, 13 provides generation examples across chain, branch and parallel instructions. Specifically, in the Chain instruction, direct generation fails to execute the third sub-task (Omitted

Table 10: Efficiency of the proposed ATA framwork.

| COMPLEXITY | 2 | 3 | ≥ 4 |
|---|---|---|---|
| DIRECT | 9.69 (s) | 14.91 (s) | 18.66 (s) |
| ATA | 10.14 (s) | 15.83 (s) | 19.95 (s) |
| TIME OVERHEAD GAIN (RELATIVE) | 4.6% | 6.2% | 6.9% |

Table 11: Generation examples of LLAMA-7B on Chain Instruction.

| | |
|---|---|
| Compositional Instruction | Blog Title: Healthy Apple Muffins
Blog Theme: Recipes and Tips for Making Delicious Apple Muffins
The above are the title and theme of the blog post we have identified. Please write an introductory paragraph of no less than 500 words to capture the reader's attention.
Please start with "Family Recipe Collection" as the introduction, then incorporate the health benefits of eating apple muffins in the middle part, and end with a rhetorical question that ties the beginning and the end together. |
| Direct Generation | Family Recipe Collection: Making and Tips for Healthy Apple Muffins ✓
In our daily lives, apples are a common and popular fruit. Not only do they taste great, but they are also rich in various nutrients such as vitamins, minerals, and dietary fiber, which are greatly beneficial to human health. Today, I will introduce you to a healthy delicacy made with apples as the main ingredient—apple muffins. They are both delicious and nutritious, making them an indispensable dish in family recipes.
The main ingredient in apple muffins is apples, and through special processing and preparation, the texture of the apples becomes more delicate and the nutrition more abundant. Additionally, apple muffins include other ingredients such as oats and honey. These additions not only enhance the taste of the muffins but also make their nutritional value more comprehensive.
So, what are the health benefits of eating apple muffins? First, apple muffins are rich in dietary fiber, which helps promote intestinal peristalsis and prevent constipation. Second, the antioxidants found in apples can effectively combat the damage caused by free radicals to the human body, slowing down the aging process. Third, the malic acid and vitamin C in apple muffins help improve skin condition and maintain skin radiance. In summary, apple muffins are a delicious and healthy food. Whether as a family recipe or a snack, they are an excellent choice. So how do you make apple muffins? Please stay tuned for our upcoming content, where we will provide detailed steps and tips for making apple muffins. Let's first take a look at the main ingredients and condiments for apple muffins:
Ingredients: Apples, oats, honey, butter
Condiments: Eggs, salt, cinnamon powder, etc. ✓
Next, please follow our subsequent content, where we will bring you the method and tips for making apple muffins. Let's get started on making this delicious and healthy apple muffin together! ✗

([**Omitted Generation**]: fail to execute the third sub-task that require to end with a rhetorical question.) |
| ATA Generation | Family Recipe Collection always has a few dishes that can evoke warm memories deep in our hearts, and apple muffins are undoubtedly one of them. Whether on the breakfast table in the morning or during afternoon tea time, apple muffins can always bring us endless satisfaction with their unique aroma and taste. Imagine a freshly baked muffin emitting an enticing apple fragrance, a gentle bite revealing a soft texture perfectly blended with the sweetness of apples, instantly awakening your taste buds. This simple yet classic delicacy is not just a treat for the taste buds but also a part of a healthy lifestyle. ✓
Apple muffins are not only delicious but also packed with nutritional value. Apples are rich in dietary fiber and vitamin C, which help promote digestion and boost the immune system. Adding apples to muffins not only increases the natural sweetness but also provides additional nutritional support for the body. Studies have shown that moderate consumption of apples helps lower cholesterol levels and improve cardiovascular health. Moreover, the antioxidants in apples help fight free radicals, slowing down the aging process. Incorporating these health benefits into muffins allows you to enjoy deliciousness while also positively impacting your body.
Making apple muffins does not require complex skills; just a few simple steps can easily be completed at home. Whether you are a baking novice or an experienced culinary expert, this recipe will allow you to easily get started. By selecting quality ingredients and mastering some small tips, you can present sweet and delicious apple muffins to your family and friends. Imagine a sunny morning, family members sitting around the dining table, enjoying the apple muffins you made by hand—that sense of happiness is irreplaceable. ✓
So, why not try making a healthy and delicious apple muffin for yourself and your family this weekend? In this fast-paced era, occasionally pausing to enjoy a simple and healthy meal may be exactly what we need. Are you ready to embark on this journey of deliciousness and health? ✓ |

Generation), which mandates that the output end with a rhetorical question. In the Branch instruction, direct generation additionally executes an unselected branch along with the selected one (Wrong Generation). In the Parallel instruction, direct generation makes mistakes in one of the questions (Mixed Generation), likely due to interference from the other question. In contrast, our proposed ATA consistently achieves precise alignment with the requirement across all three instructions, showing its clear superiority.

Table 12: Generation examples of LLAMA-7B on Branch Instruction.

| | |
|---|---|
| Compositional Instruction | Hu Juren was a Chinese writer and educator, renowned for his literary works and contributions to education. His works span multiple fields, showcasing a profound insight into society and human nature.
- If the above content mentions Hu Juren and references his work "Self-Encouragement Couplets," please provide the full text of "Self-Encouragement Couplets" and analyze each sentence point by point. Otherwise, provide a biography of Hu Juren in no more than 300 words.
- If the above content does not mention Hu Juren but references qualities such as perseverance and persistence, please expand on this content to about 500 words using only commas and periods, and conclude with a famous quote. Otherwise, state that the article does not contain relevant content. |
| Direct Generation | The content above does not mention Hu Juren. If it did mention qualities like perseverance and persistence, please expand on this content to about 500 words, using only commas and periods, and conclude with a famous saying. Otherwise, please state that the article does not contain relevant content.
Hu Juren, a Chinese writer and educator, dedicated his life to the relentless pursuit of knowledge and the exploration of truth. He firmly believed that only through perseverance and unwavering determination could one achieve success in life.
He once said, "Life would be perfect if it were as it was at first sight, why lament the autumn wind's sorrowful fan? People's hearts change easily, yet they say it is the heart that is fickle." This quote reminds us that life is fleeting, and we must cherish every moment, not letting time slip away in vain. At the same time, we must be brave enough to face changes, constantly adjusting our mindset to adapt to an ever-changing environment. ✓
Hu Juren's persistence and dedication ultimately led him to achieve remarkable success in the fields of literature and education. He always believed that only by putting in immense effort could one attain true success. This spirit is something we should all strive to emulate.
Therefore, we should be like Hu Juren, embodying the qualities of perseverance and persistence, to achieve success on the path of life. As the ancients said, "If you carve and give up, the rotten wood will not break; if you carve and do not give up, the metal and stone can be engraved." As long as we persist, we will surely achieve success.
In the journey of life, we will all encounter setbacks and difficulties, but as long as we can persevere and persist, we will surely overcome the difficulties and achieve success. As Hu Juren said, "Life would be perfect if it were as it was at first sight, why lament the autumn wind's sorrowful fan? People's hearts change easily, yet they say it is the heart that is fickle." We must learn to persist, not be defeated by difficulties, face life courageously, and pursue our dreams.
Finally, let us end with a famous saying, "Life is like a play, and everything can be endured." May we persevere and forge ahead on the path of life, achieving our own success. ✗

(**[Wrong Generation]**: Erroneously execute the unselected inactive branch.) |
| ATA Generation | The aforementioned content does not specifically mention Hu Juren or the content of his "Self-Encouragement Couplet." Therefore, I will provide a brief introduction to Hu Juren's life.
Hu Juren (1434-1484), courtesy name Shuxin, and sobriquet Jingzhai, was a renowned Neo-Confucian scholar and educator of the Ming Dynasty. He was born in Jishui County, Jiangxi Province, and showed remarkable intelligence and a love for learning from a young age. He later studied under the famous Neo-Confucian scholar Wu Yubi, whose teachings profoundly influenced him. Throughout his life, Hu Juren dedicated himself to education, founding academies in his hometown and nurturing a large number of outstanding talents. His educational philosophy emphasized moral cultivation and character development, advocating for "learning for practical application," which had a lasting impact on future generations. Hu Juren's literary works also hold significant influence, often reflecting his deep insights into society and human nature. His writings, including the "Collected Works of Jingzhai," showcase his exceptional contributions to both literature and education. ✓ |

# D  THE USE OF LLMS

In this section, we clarify that no LLMs were employed in the writing or polishing of this paper. All content presented herein is the result of original research and critical evaluation by the authors.

Table 13: Generation examples of LLAMA-7B on Parallel Instruction.

| Compositional Instruction | 1. Terry sells 6 milkshakes for $5.50 each, nine burger platters for $11 each, and 20 sodas for $1.50 each. How much money does he make in total?
2. Paul is at a train station and is waiting for his train. He isn't sure how long he needs to wait, but he knows that the fourth train scheduled to arrive at the station is the one he needs to get on. The first train is scheduled to arrive in 10 minutes, and this train will stay in the station for 20 minutes. The second train is to arrive half an hour after the first train leaves the station, and this second train will stay in the station for a quarter of the amount of time that the first train stayed in the station. The third train is to arrive an hour after the second train leaves the station, and this third train is to leave the station immediately after it arrives. The fourth train will arrive 20 minutes after the third train leaves, and this is the train Paul will board. In total, how long, in minutes, will Paul wait for his train? |
|---|---|
| Direct Generation | Here are the answers:
[Answer 1]: Terry makes a total of:
6 milkshakes * $5.50 = $33.00,
9 burger platters * $11.00 = $99.00,
20 sodas * $1.50 = $30.00,
Total: $33.00 + $99.00 + $30.00 = $162.00 ✓
[Answer 2]: Paul will wait for his train for:
10 minutes (first train arrives), 20 minutes (first train stays), 10 minutes (second train arrives), 10 minutes (second train stays), 30 minutes (second train leaves, third train arrives), 0 minutes (third train stays), 20 minutes (third train leaves, fourth train arrives),
Total: 10 + 20 + 10 + 10 + 30 + 0 + 20 = 100 minutes ✗

**[Mixed Generation]: Impacted by potential interference from the information in question 1.** |
| ATA Generation | Here are the answers:
[Answer 1]: Terry makes a total of:
6 milkshakes x $5.50 = $33.00,
9 burger platters x $11.00 = $99.00,
20 sodas x $1.50 = $30.00,
Total: $33.00 + $99.00 + $30.00 = $162.00 ✓
[Answer 2]: Paul will wait for his train for:
First train: 10 minutes, First train stays in the station for 20 minutes, so Paul has to wait another 20 minutes after the first train leaves.
Total wait time for the first train: 10 + 20 = 30 minutes,
Second train arrives 30 minutes after the first train, and stays in the station for 20/4 = 5 minutes. Paul has to wait 5 minutes after the second train leaves.
Total wait time for the second train: 30 + 5 = 35 minutes,
Third train arrives 60 minutes after the second train, and leaves immediately. Paul has to wait 0 minutes after the third train leaves.
Total wait time for the third train: 60 minutes,
Fourth train arrives 20 minutes after the third train leaves, which is the train Paul will board. Total wait time for the fourth train: 20 minutes,
Total wait time: 30 + 35 + 60 + 20 = 145 minutes ✓ |

