# OpenReview forum: "Attend to the Active: Structure-Aware Dynamic Attention in LLMs for Compositional Instruction Following"
_ICLR.cc/2026/Conference — ICLR 2026 Poster_

### Official Review · Reviewer_Hrq3 · 2025-10-30

**Soundness:** 2
**Presentation:** 2
**Contribution:** 3
**Rating:** 6
**Confidence:** 4

**Summary:**

This research studies the complex instruction-following problem. The authors consider that there are three structures of sub-tasks from the user instructions, and propose to manipulate the attention-patterns for different task structures such that the model can attend to the correct contents for each sub-task. The results of the control experiments support the effectiveness of the proposed method.

**Strengths:**

1. The overall idea of manipulating the attention mask to enforce the model to attend to certain sub-tasks for complex instruction following is reasonable.
2. The comprehensive ablation study provides evidence for the effectiveness of each proposed component.
3. For most of the manuscript, the writing is clear.

**Weaknesses:**

1. The authors decompose sub-tasks by leveraging an LLM, which can be a significant bottleneck to the framework when it tries to handle more challenging tasks. Although the performance of structure identification (Appendix B.1) looks good on current datasets, I am not sure whether this reliability can extend to more challenging tasks/datasets, such as AIME 24/25 for mathematical solving. The reason I raise this concern is that, for many hard questions, the sub-tasks are not explicitly presented in the user instructions, so the model needs to plan them by itself at first.

2. Figure 3 has not been rendered properly.

3. How is the model's performance on general (simple) instructions, such as email drafting and poem writing? In other words, whether the system is robust to the instructions that do not hold the hypothesis that they have multiple sub-tasks?

4. How much is the overall overhead for model inference, including the time of annotating sub-task structures?

**Questions:**

See Weakness.

---

> ### Author Response · Authors · 2025-11-23
> **Author Response to Reviewer Hrq3 (PART I)**
>
> Thank you for your thoughtful and constructive feedback. We appreciate your recognition of our work’s **reasonableness, clarity, and effectiveness**. Below, we provide detailed responses to your comments and hope our clarifications address all your concerns satisfactorily.
>
> ---
>
> **Q1: Effectiveness on more challenging tasks.**
>
> Thanks for your insightful suggestion.
>
> We first clarify that ATA is tailored for complex compositional instructions with explicitly structured sub-tasks. For tasks with implicit structures, ATA operates similarly to a direct method without attention steering (as noted in response to Q2).
> However, the key insight of ATA is to ensure **faithful execution of an intended sub-task** by reducing attentional distraction. This **naturally complements** existing techniques like **planning** and **decomposition**, which **uncover implicit structure** by breaking complex tasks into simpler, clear sub-tasks. Once the structure is identifiable, ATA can **serve as a post-hoc module** to further improve attention adherence and reasoning fidelity.
>
> To validate this, we integrate ATA with advanced structure-inducing methods and evaluate it on challenging GSM8K and HumanEval benchmarks:
>
> |Method|GSM8K|HumanEval|
> |---|---|--|
> | Direct I/O| 77.5 | 67.8 |
> | Decomposition [1]| 79.6 | 68.3 |
> | Decomposition + ATA (ours) | **81.0**|**68.8**|
> | Plan-and-Solve [2]| 81.2 | 69.4 |
> | Plan-and-Solve + ATA (ours)| **82.7**|**70.9**|
>
> Results show that ATA **consistently boosts performance** of baseline methods, serving as a robust **general-purpose attention control module**. While structure identification remains a bottleneck, ATA **effectively generalizes to complex scenarios** when combined with advanced structure-inducing techniques, further improving the model's focus.
>
> We will include the discussion in our revision and explore structure-agnostic attention steering mechanism in future work to further broaden our applicability.
>
> > [1] Wen, Bosi, et al. "Benchmarking complex instruction-following with multiple constraints composition." Advances in Neural Information Processing Systems 37 (2024): 137610-137645.\
> > [2] Wang, Lei, et al. "Plan-and-Solve Prompting: Improving Zero-Shot Chain-of-Thought Reasoning by Large Language Models." ACL 2023.
>
> ---
>
> **Q2. Figure 3 has not been rendered properly.**
>
> Thanks for pointing this out. We will carefully polish this figure in the revision.
>
> ---
>
> **Q3: Robustness to general instruction dataset.**
>
> Thanks for your valuable suggestion.
>
> To demonstrate ATA's robustness on general-purpose tasks, we evaluate it on the general instruction following benchmark **IFEval** [1], which contains email drafting or poem writing tasks. The results are presented below:
>
> |Method| IFEval |
> |---|---|
> | Direct I/O  |73.5 |
> | CoT Prompting |71.4|
> | Think-Execute |74.6|
> | ATA (ours) |74.2 |
> | **Think-Execute + ATA (ours)**| **75.7** |
>
> We first acknowledge that IFEval's instructions primarily involve ***multiple entangled constraints within a single instruction***, rather than the ***explicit compositional sub-task structure ATA is designed for***.
> In such scenarios:
> - **`Direct Application`**: ATA operates like a direct method without attention steering, **yielding only marginal improvements**. This highlights ATA's **inherent safety**: when no clear structure is detected, it avoids applying unnecessary masking that could degrade performance.
> - **`With Think-Execute`**: When combined with the planning-based method Think-Execute, which uncovers implicitly sub-task structures, ATA **further boosts model performance**, showcasing its **robustness as a general attention control module** for enhancing existing techniques by steering model focus over the planned steps.
>
> Overall, ATA is both safe for simple instructions and effective in enhancing complex tasks, making it a robust tool for improving instruction-following in real-world scenarios.
> We will include this discussion in the revision.
>
> > [1] Zhou, Jeffrey, et al. "Instruction-following evaluation for large language models." arXiv preprint 2023.
>
> ---

---

> ### Author Response · Authors · 2025-11-23
> **Author Response to Reviewer Hrq3 (PART II)**
>
> **Q4: Time overhead for model inference.**
>
> The overhead of ATA is minimal and arises from two sources:
> 1.  **`ATA mechanism during Inference:`** As detailed in **Table 9 (Appendix [B.3])**, the ATA mechanism itself, which involves calculating attention entropy and applying masking, is **computationally lightweight**, incuring a negligible overhead of **less than 7%** per request.
> 2.  **`Structure Identification:`** This is a **one-time, pre-processing step** performed by an LLM. While it has a relatively higher time overhead, as summarized in the table below, it can be cached, allowing the cost to be **amortized across repeated queries**.
>
> |Complexity| 2| 3 | ≥ 4 |
> |--|----|----|----|
> |Direct I/O | 9.69 (s)|14.91 (s) |18.66 (s)|
> | ATA| 10.14 (s) |15.83 (s) |19.95 (s)  |
> | Structure Identification|1.28 (s)| 2.46 (s)|4.37 (s)|
>
> We will include the summarized table in the appendix to provide full transparency on the practical costs.

---

### Official Review · Reviewer_grgz · 2025-10-30

**Soundness:** 3
**Presentation:** 2
**Contribution:** 2
**Rating:** 4
**Confidence:** 4

**Summary:**

This paper proposes an attention steering technique for improving instruction following by LLMs for compositional instructions. Three compositions are considered: chaining (sequential), branching (conditional), and parallel execution of sub-tasks. The key claim is that at any generation step, only one sub-task should be active irrespective of the type of composition. The steering technique, Attend To the Active (ATA), first requires the composite instruction to be distilled to the composition type and list of sub-tasks. It then first constructs a mutual attention masking  matrix that downweighs attention between independent tasks, and a dynamic attention masking matrix that downweighs attention to tasks independent of the current "active task" (the active task is identified as the one receiving the most attention at the current step of generation). These masks are applied at every step of generation to a selected subset of attention heads. The experiments show that ATA outperforms other instruction following techniques and attention steering methods on instructions from the three composition types. An analysis of components of ATA demonstrates that it is robust to how the structure information is presented and the masking degree, but the structure information along with the masking modules and the control strategy (identification of the active task) are essential -- the selection of attention heads is crucial to the success of ATA.

**Strengths:**

- The paper is well written and the key ideas are fairly easy to understand.

- The problem of better instruction-following with compositional instructions for LLMs is important and the proposed approach, ATA, of masking attention on sub-tasks based on structural exclusiveness and the current active task seems novel.

- The experiments include a good variety of baselines (other instruction following prompting strategies and attention steering methods) and provide insights into how various components of ATA are useful.

Overall, I think that the paper studies an important problem and the experiments demonstrate effectiveness over other techniques.

**Weaknesses:**

- ATA essentially ensures that only one sub-task is attended to at any point but does not focus on whether all sub-tasks are eventually attended to (for chaining / parallel compositions). Improved instruction following would require that the order of the tasks is also taken into account (for chaining), and that all sub-tasks are "accomplished" (for parallel). At the least, I would expect this to be explicity mentioned in the text and some analysis from the experiments on whether this happens with ATA.
- For branch compositions, I would also expect the condition to be a sub-task that is attended to first. But the prompt template in Section 4.1 does not seem to include the condition during structure identification. Moreover, as I mention in the previous point, ATA does not take into account that only one of the sub-tasks should be accomplished (at least in the way it is defined and discussed in Section 4.2).
- I find it difficult to understand which tokens are being included in the identification of the current active sub-task. For Equation 5, it seems like it computes the average attention between the tokens of a task instruction T_i (k) and all tokens after the instruction (q >= k)? Would this not include other tokens of T_i, othe following sub-tasks, and the entire generation sequence so far? The preceeding description of the score seems to convey that the score is computed with respect to the next token at a given step (lines 283-286).
- The identification of the active task also mentions the entropy threshold. How this threshold is computed and how the value affects the performance is not discussed in the main text / experiments. Moreover, I do not understand what happens is the entropy is over the threshold.

Overall, I think this work would benefit from providing more details on the technique (the active sub-task score, the entropy threshold, the inclusion of the condition for branching tasks) and the contribution can be improved by taking the overall structure into account (sequential tasks should happen in order, parallel tasks should all eventually happen, etc.).

**Questions:**

I will summarize my questions from the weaknesses section above (please refer to that section for more details):
1. Which tokens are included as "next tokens" in the score computation for the active task?
2. How is the entropy threshold computed and how sensitive is ATA to this choice?
3. How is the condition for branching tasks factored in during structure identification?
4. Does ATA promote global structure following by restricting attention to the current active task -- do sequential / parallel sub-tasks all eventually happen?

Additionally:

5. How does the structure identification work for the nesting experiments (Table 4)? What does the prompt look like here?
6. In the ablation on structure identification (Figure 4(b)), what do the variants partial, original and human-revised mean?
7. For active sub-task identification, what happens when the entropy is over the threshold?

---

> ### Author Response · Authors · 2025-11-23
> **Author Response to Reviewer grgz (PART I)**
>
> Thank you for your careful and insightful feedback. We appreciate your recognition of our work’s **clarity, novelty, insightfulness, and effectiveness**. Below, we provide responses to each of your questions and hope our clarifications address all your concerns satisfactorily.
>
> ---
>
> **Q1: Analysis of the global structure following in ATA.**
>
> Thank you for your insightful question.
>
> We recognize the importance of taking the overall structure into account for effective global instruction following. Actually, ATA addresses this through an **"Active Control" strategy**, ensuring valid active sub-task identification by considering the global structure:
> - **`For Chaining / Parallel tasks`:** Enforces **strict sequential progression** (1 → 2 → 3...) for active sub-task identification, **preventing skips or revisits**. While we do not impose a hard constraint to eventually "accomplish" all sub-tasks, this sequential progression mechanism strongly **encourages task completion**.
> - **`For Branching tasks`:** Once a branch is selected, enforces **consistent active sub-task during generation**, preventing switching or attending to other branches, thereby ensuring **faithful execution of the chosen sub-task**.
>
> Detailed analysis of this strategy is discussed in ***Section 5.3***, and the ***ablation study in Table 2 (- w/o Active Control)*** highlights its effectiveness in **reducing occasional identification errors** and **promoting global instruction following**.
>
> ---
>
> **Q2: How is the condition for branching tasks factored in during structure identification?**
>
> During structure identification for branching instructions like "If A, then B, else C," our module is designed to **only extract the executable text ("B" and "C") as subtasks**, intentionally **leaving the condition ("A") as part of the shared global context**. This design is vital for safe comprehension since **conditions ("A") are for *reasoning*, not *generation*.**
>
> Specifically, the condition "A" guides the model’s reasoning by comprehending the entire logic and forming necessary connections between "A" and its potential consequences ("B" and "C"). **Isolating "A" as a separate sub-task would force the model to process the condition before "seeing" the outcomes**, disrupting overall comprehension.
>
> Instead, our strategy allows the model to process the global context, including "A," while ATA masks mutual attention between exclusive branches ("B" and "C") to ensure disentangled comprehension. Once a branch is selected, ATA focuses attention on the chosen sub-task, avoiding interference from the unselected branch.
>
> We will clarify this process and its rationale in the revised manuscript.
>
> ---
>
> **Q3: Clarification about the "next tokens" for score computation in active task identification.**
>
> Thank you for this detailed question.
>
> For active task identification, our goal at each generation step `t` is to identify which sub-task **dominates the model's focus and guides the prediction of next token (`t+1`)**.
> Since we cannot measure attention from a token that has not yet been generated, we evaluate each sub-task's **influence on the generation process up to current step `t`**. The sub-task that attracts the highest cumulative attention so far is deemed to contribute most significantly to the next token prediction.
>
> As formalized in Equation 5, the score for a subtask `T_i` at step `t` is the **average attention received by all its tokens `k` from subsequent tokens `q` (up to step `t`)**, i.e., `k <= q <= t`. Importantly, the score **actually excludes attention from tokens of other following sub-tasks `T_j`** as their mutual attention is masked prior to generation, which is a core part of our method.
>
> We will revise the manuscript to clarify this and provide a more detailed annotation of Equation 5 to prevent ambiguity.

---

> ### Author Response · Authors · 2025-11-23
> **Author Response to Reviewer grgz (PART II)**
>
> **Q4: Details and analyses about the entropy threshold.**
>
> Thanks for your careful comments. We provide details about the entropy threshold for active sub-task identification point-by-point below:
>
> -   **`Entropy Threshold Computation`:** To adapt to instruction complexity, the entropy threshold is set **proportionally to the number of sub-tasks N** rather than a fixed value. Specifically, it is dynamically computed as `α * log(N)`, where `N` is the number of subtasks and `α` is a hyperparameter (empirically set to 0.5).
> This is because the **upper bound of entropy** for an N-outcome distribution is `log(N)`, which corresponds to a uniform distribution and represents maximum uncertainty.
>
> -   **`Sensitivity to Entropy Threshold`:** We conduct a sensitivity analysis to this choice and the results are shown below:
>
> | Entropy Threshold Setting | Chain | Branch | Parallel |
> | :--- | :---: | :---: | :---: |
> | Dynamic (α=0.3) | 62.1 | 58.2 | 69.3 |
> | **Dynamic (α=0.5)** | **63.0** | **58.7** | **69.9** |
> | Dynamic (α=0.8) | 61.7 | 56.8 | 68.4 |
> | Static ($\epsilon$=0.2)  | 61.4 |57.6 |67.2|
>
> The results showcase that our dynamic entropy threshold is **relatively robust and outperforms static thresholds**.
>
> -  **`When entropy is over the threshold`:** If the attention score entropy exceeds the threshold, it indicates the model's focus is **not sharply concentrated** on any single sub-task. In this case, our system acts as a **safety mechanism**: it **maintains the previous active sub-task identification state** (either no masking or the previously active masking). This prevents erroneous changes when the model has not made a determined decision or legitimately considers the global context, ensuring valid active sub-task identification.
>
> We will make this clearer in the revision.
>
> ---
>
> **Q5: Prompt for nested structure identification.**
>
> To handle nested structures, we design a prompt that explicitly guides the LLM to perform a **recursive identification** of the instruction structure, which is built on three key principles to ensure high accuracy:
>
> 1. **Explicit Instruction for Recursion:** The prompt ***explicitly states that a sub-task can itself be a compositional task*** (Chain, Branch, or Parallel) and must be analyzed at all levels.
> 2.  **Strict Output Formatting:** It ***defines the recursive JSON format*** required for nested tasks.
> 3.  **In-Context Examples:** We ***provide high-quality examples of nested structures*** and their corresponding JSON outputs, enabling powerful in-context learning even with moderately-sized models.
>
> We will provide the complete prompt in the appendix in our revised manuscript. The core of the prompt is included below for your reference:
>
> > You are an excellent logic expert. Given a compositional task and the definition of structure types...
> >
> > \### **Definitions**
> > 1.  **Structure Types:**
> >     *   **Chain:** ...
> >     *   **Branch:** ...
> >     *   **Parallel:** ...
> > 2.  **IMPORTANT: Analyze for Nested Structures.** A sub-task can itself be a compositional task (Chain, Branch, or Parallel). You must analyze the structure at all levels.
> >
> > \### **Output Requirements**
> > *   **Format:** Output a single JSON object:
> > ```json
> > {
> >   "Structure Type": "Chain/Parallel/Branch",
> >   "Sub-tasks": [
> >     "Sub-task1",
> >     {
> >       "Structure Type": "Chain/Parallel/Branch",
> >       "Sub-tasks": [
> >         "Sub-task2.1",
> >         "Sub-task2.1"
> >       ]
> >     },
> >    "Sub-task3"
> >   ]
> > }
> > ```
> > *   **Content:** The `Structure Type` must be one of "Chain", "Branch", or "Parallel". A sub-task can be a `string` (for a simple subtask) or a nested `JSON object` (for a nested subtask).
> > *   **Accuracy:** The extrated sub-tasks must be **exact, verbatim excerpts** from the original text. Do not modify, delete, or add any words.
> >
> > \### **Example of a Nested Task**
> >
> > **Task: ...**
> >
> > **Output: ...**
> >
> > ---
> > Now, apply this exact methodology to the following task.
> >
> > **Task:** {input task}
>
> ---
>
> **Q6: Clarification about variants naming in Figure 4(b).**
>
> The detailed definitions for the legends in Figure 4(b) are as follows:
>
> -   **`human-revised`:** This refers to the **ground-truth structure information**, which has been manually annotated or corrected by humans. It serves as an upper bound on performance with perfect structure identification.
> -   **`original`:** This is the direct, **raw output** from our LLM-based structure identification module, used as our main method.
> -   **`partial`:** This is a control variant where the structure information is **intentionally degraded** (e.g., only part of the sub-tasks are identified) to test ATA's robustness to imperfect structure identification.
>
> We will clarify this in the revision.
>
> ---

---

> > ### Comment · Reviewer_grgz · 2025-11-26
> >
> > Thank you so much for the detailed response. I appreciate the clarifications on the branching condition, the next tokens, the entropy threshold, and the nested instructions.
> >
> > I am still confused about the claim in response to Q1 that ATA "enforces strict sequential progression (1 → 2 → 3...) for active sub-task identification, preventing skips or revisits". If I understand this correctly, the active subtask is chosen as the one receiving the maximum average attention from future tokens. Is the sub task sequence being strictly enforced during the dynamic attention masking or does this assume that the generation is going to naturally attend more to subtasks in that order?

---

> > > ### Author Response · Authors · 2025-11-27
> > > **Further clarification about the "strict sequential progression" for active sub-task identification**
> > >
> > > We are glad our previous clarification addressed many of your concerns and appreciate the opportunity to further clarify the "strict sequential progression" for active sub-task identification.
> > >
> > > To clarify, ATA’s active sub-task identification is **always based on the highest score**, a sub-task cannot be recognized as active without achieving the highest score.
> > > However, to **ensure robust and safe progression** aligned with the instruction's global structure, ATA **incorporates an active control mechanism** on top of the score-based identification, avoiding confusion or errors caused by the model’s occasional attention shifts.
> > >
> > > Specifically, for sequential/parallel instructions, the natural and logical order of execution should start with the first sub-task (T1) and progress sequentially (T1->T2->T3->...):
> > > 1. **`Initial Identification`**: At the start of generation, **only T1 can be marked active when it achieves the highest score**. If T2 or T3 temporarily dominates early attention, no sub-task will be marked active, and attention steering will not be applied.
> > > 2. **`Controlled Progression`**: During generation, the current active sub-task (T_i) **only switches to the sequential sub-task T_i+1 when T_i+1 achieves the highest score**. If other sub-tasks (e.g., T_i-1 or T_i+2) dominate temporarily, ATA retains T_i as active, preventing premature or illogical transitions.
> > >
> > > This ensures active sub-task **switching occurs naturally and logically** (e.g., T_i → T_i+1), avoiding interference from attention spikes to skipped or revisited sub-tasks, and **ensuring safe and consistent execution** aligned with the instruction’s global structure.
> > > The effectiveness of this control mechanism is demonstrated in the ablation study in Table 2 (- w/o Active Control). Results show that ATA improves task progression and reasoning robustness with this control.
> > >
> > > We hope this explanation resolves your confusion, and we will refine the manuscript to clearly explain this mechanism. Thank you again for raising this question!

---

> > > > ### Comment · Reviewer_grgz · 2025-11-27
> > > >
> > > > Thank you so much for the clarification!
> > > >
> > > > I assumed that active sub task selection and the active control strategy were synonymous, which led to this confusion. In order to help other readers like me, I would encourage the authors to also coin the term and briefly discuss/highlight it in Section 4 (I see it first mentioned in Section 5.3).
> > > >
> > > > I am increasing my score since the authors have addressed my concerns.

---

> > > > > ### Author Response · Authors · 2025-12-01
> > > > > **Thanks for kindly increasing your score.**
> > > > >
> > > > > We're delighted to hear that your concerns have been resolved, and we greatly appreciate the increase of your score.
> > > > > We will make sure to incorporate the points discussed into the final revision.
> > > > >
> > > > > Thank you again for your positive feedback and kind recognition.

---

### Official Review · Reviewer_TRL7 · 2025-10-31

**Soundness:** 3
**Presentation:** 4
**Contribution:** 4
**Rating:** 8
**Confidence:** 4

**Summary:**

This paper improves the instruction following behavior of LLMs by identifying three different compositional structures of instructions which LLMs struggle to follow. They propose an attention modification method which ensures that the LLM only attends to the relevant sub-instruction at any time during generation, and they show that this improves instruction following.

**Strengths:**

* The paper is very well written and easy to follow.
* The approach is novel and addressed an important problem with a relatively lightweight method.
* A large number of baselines and models are used in the evaluation which shows that the proposed approach is most effective.
* The attention head selection strategy proposed in Appendix A.3 is shown to be effective while using substantially less compute than baselines which makes these methods easier to use.
* Results on nested composition also show that the method remains effective.

**Weaknesses:**

* Missing standard deviations for results.
* The datasets used for evaluation are designed to have the three types of composition, but a traditional instruction following dataset with more “real” tasks is not used for evaluation.

**Questions:**

* Figure 4a shows that ATA reduces the identified generation errors, but it doesn’t go to 0. Why do you think some of these errors remain? Are the wrong subtasks selected by the model’s attention?
* How often is active subtask selection incorrect?

---

> ### Author Response · Authors · 2025-11-23
> **Author Response to Reviewer TRL7**
>
> Thank you for your valuable feedback and insightful comments. We sincerely appreciate your recognition of the **novelty, clarity, and effectiveness** of our work. In the following, we provide detailed responses to each of your questions, and we hope our clarifications are helpful to resolve all your concerns.
>
> ---
>
> **Q1: Missing standard deviations.**
>
> Thank you for pointing this out. To improve the rigor of our results, we will **add standard deviations to our main results** in the appendix of our revised manuscript.
>
> ---
>
> **Q2: Evaluation on traditional instruction following dataset with general tasks.**
>
> Thanks for your valuable suggestion.
>
> To demonstrate ATA's robustness on general-purpose tasks, we additionally evaluate it on the traditional instruction following benchmark **IFEval** [1]. The results are presented below:
>
> | Method| IFEval |
> |-------|-------|
> | Direct I/O  | 73.5 |
> | CoT Prompting | 71.4  |
> | Think-Execute |  74.6  |
> | ATA (ours) | 74.2 |
> | **Think-Execute + ATA (ours)**  | **75.7** |
>
> We first acknowledge that IFEval's instructions primarily involve ***multiple entangled constraints within a single instruction***, rather than the ***explicit compositional sub-task structure ATA is designed for***.
> In such scenarios:
> - **`Direct Application`**: ATA operates like a direct method without attention steering, yielding only marginal improvements. This highlights ATA's **inherent safety**: when no clear structure is detected, it avoids applying unnecessary masking that could degrade performance.
> - **`With Think-Execute`**: When combined with the planning-based method Think-Execute, which uncovers implicitly sub-task structures, ATA **further boosts model performance**, showcasing its **robustness as a general attention control module** for enhancing existing techniques by steering model focus over the planned steps.
>
> Overall, ATA is both safe for simple instructions and effective in enhancing complex tasks, making it a robust tool for improving instruction-following in real-world scenarios.
> We will include this discussion in the revision.
>
> > [1] Zhou, Jeffrey, et al. "Instruction-following evaluation for large language models." arXiv preprint 2023.
>
> ---
>
> **Q3: Analysis of remaining generation errors.**
>
> Thanks for your careful comment.
>
> The fundamental reason that generation errors are not completely eliminated is that ATA is designed to **mitigate attentional distraction during comprehension and generation**, rather than to **correct fundamental flaws in the model's inherent decision-making or reasoning abilities.**
>
> Specifically, ATA eliminates comprehension confusion by minimizing mutual attention between sub-tasks, and enables the model to focus solely on the *chosen* sub-task so that can execute its plan faithfully. However, if the model makes an inherently wrong decision or reasoning error (e.g., selecting the wrong branch in a conditional statement), ATA cannot rectify such mistake through attention steering, leading to the persistence of generation errors.
>
> ---
>
> **Q4: How often is active sub-task selection incorrect?**
>
> To directly assess ATA's precision in identifying active sub-tasks for effective attention steering, we measure its **Attention Consistency**, defined as the percentage of decoding steps where the identified active subtask matches the ground-truth.
> As shown below, ATA is **highly effective at dynamically aligning the model's focus with the intended execution path**:
>
> | Metric | Chain | Branch | Parallel |
> | :--- | :---: | :---: | :---: |
> | Attention Consistency | 81.2% | 75.1% | 86.4% |
>
> As for the incorrect active sub-task selection, it can be analyzed in two modes:
>
> 1.  **`Identification Failure:`** This occurs when ATA fails to track the model's actual focus. However, this is **extremely rare**, as our active sub-task identification mechanism achieves **over 98% accuracy** in correctly tracking the model's focus, confirming its reliability.
>
> 2.  **`Model Reasoning Failure:`** This is the **primary source of error** and relates to our response to Q3. In this case, the model itself focuses on the incorrect sub-task, and while our identification module detects it accurately and ATA steers attention accordingly, the attended sub-task remains incorrect **due to the model's inherent flaw decision**.
>
> In essence, ATA aids comprehension by eliminating confusion and enabling faithful generation by enhancing focus, but it cannot rectify inherent reasoning flaws.
> This limitation is tied to the model's inherent capabilities, which ATA is not designed for. We will include this discussion in the revision.

---

### Official Review · Reviewer_qkLE · 2025-11-01

**Soundness:** 3
**Presentation:** 3
**Contribution:** 3
**Rating:** 8
**Confidence:** 4

**Summary:**

The paper proposes Attend to the Active (ATA), an inference-time method that steers a model’s attention toward the currently “active” sub-task within a complex instruction. ATA masks attention between mutually exclusive sub-tasks and then boosts attention to the detected active sub-task. The authors evaluate on chain, branch, and a synthetic parallel setup and report consistent gains over prompting and planning baselines, with ablations on masking strength and head selection. They also include robustness checks for structure identification quality and some results on nested compositions.

**Strengths:**

1. Good idea and implementation. The focus on compositional instruction following is timely. Doing it with attention masking at inference is interesting because it avoids retraining and aims to localize interference between sub-tasks.

2. Clear writing and organization. The paper explains the three structure types, gives concrete examples, and walks through the masking and steering pipeline in a way that is easy to follow.

3. Strong experiments and analysis. The experiments cover two base models, three composition types, and include ablations for masking degree and number of steered heads, as well as component removals. The figures show sensitivity to α and head counts, and tables report the impact of removing structure info, mutual masks, dynamic masks, and active control.

**Weaknesses:**

1. Parallel instruction data is simplistic - The “parallel” benchmark is built by concatenating independent GSM8K problems. This construction makes sub-tasks independent by design, which favors attention masking because there is little need for cross-task sharing or entity linking. A harder parallel set, closer to the realistic prompts illustrated in figure 2, would be better test.

2. General capability preservation is under-tested - Because ATA modifies attention patterns, it is important to show that language abilities are not degraded. Following evaluations in [1] it would be good to measure side effects on general text quality.


References:
[1] Stolfo et al., “Improving Instruction-Following in Language Models through Activation Steering,” ICLR 2025.

**Questions:**

None.

---

> ### Author Response · Authors · 2025-11-23
> **Author Response to Reviewer qkLE**
>
> Thank you for your insightful review and positive feedback. We are particularly encouraged by your recognition of our work's **timely motivation**, **interesting implementation**, and **strong analysis**. Below, we address your valuable suggestions point-by-point and hope to address all your concerns.
>
> ---
>
> **Q1: Evaluation on a harder parallel instruction dataset.**
>
> Thank you for this valuable suggestion.
>
> To demonstrate the practical utility of ATA on more realistic parallel tasks, we curate **Parallel-Challenge**, a new dataset derived from ComplexBench, considering the lack of public datasets for challenging parallel instructions. This dataset comprises approximately 100 parallel tasks carefully selected, rewritten by GPT-4o, and validated by human experts to ***include cross-task sharing constraints***.
>
> As shown in the table below, ATA **maintains a significant performance advantage**, confirming its **effectiveness even in more demanding, realistic scenarios**.
>
> | Method | Parallel-Challenge |
> | :--- | :---: |
> | Direct I/O | 54.72 |
> | CoT Prompting | 58.24 |
> | Decomposition | 55.81 |
> | Think-Execute | 57.16   |
> | **ATA (ours)** | **61.37** |
>
> We will include a detailed description of this benchmark, along with results and analysis, in the appendix of our revised manuscript. Below are **two examples** from the **Parallel-Challenge** dataset:
>
> > **Example 1: Shared Constraints**
> > Based on the content of the Yelp reviews below, predict the rating each author would give this restaurant (Negative, Neutral, Positive). Please remain neutral and conduct the analysis without emotion. Use inferential language, incorporating reasoning words such as "because" and "therefore".
> > """ Review 1: ... """
> > """ Review 2: ... """
> >\
> > *This task is challenging because the model must independently apply the same complex set of stylistic and reasoning constraints ("neutral," "inferential," "use 'because'") to two different pieces of text.*
>
> > **Example 2: Shared Conditions**
> > Analyze the following artists and their representative works:
> > 1. Michelangelo — *The Bronze Age*.
> > 2. Leonardo da Vinci — *Mona Lisa*.
> > When the artist and the work match, generate a biography of the artist, limited to around 50 characters. If they do not match, output "Please re-enter".
> >
> > *In this case, the model must follow the same conditional logic for two separate items, requiring it to maintain focus on one item while avoiding interference from the other.*
>
> ---
>
> **Q2: Evaluation of general capability preservation.**
>
> Thanks for your insightful suggestion.
>
> To evaluate ATA's general response quality and assess whether it compromises the model's fundamental language abilities, we **perform a GPT-4o-based evaluation** on our benchmarks **following the methodology in [1]**. Specifically, we extract base query for each instruction and generate corresponding scoring questions to **measure general response quality** (details in [1]).
>
> The table below shows ***the average differences in response quality scores relative to a simple Baseline (Base Query)***, isolating the impact of adding complex instructions and ATA's attention masking.
>
> | Setting | Chain | Branch | Parallel |
> | :--- | :---: | :---: | :---: |
> | Direct I/O (Base Query) | 0 | 0 | 0 |
> | Direct I/O (w/ Instruction) | -0.17 | -0.41 | +0.06 |
> | **ATA (w/ Instruction)** | **-0.21** | **-0.28** | **+0.09** |
>
> While adding complex instructions can degrade the general response quality as adherence to constraints may impact response fluidity, ATA shows a robust performance:
>
> - **Chain**: ATA introduces **negligible additional quality drop** (-0.17 to -0.21), confirming its safety.
> - **Branch**: ATA **alleviates the side-effect** (-0.41 to -0.28) by masking irrelevant branches, reducing attentional confusion and enabling more focused and coherent responses to the selected branch.
> - **Parallel**: ATA **improves response quality** (+0.06 to +0.09) by reducing interference between independent sub-tasks, resulting in cleaner execution and higher overall output quality.
>
> This demonstrates that ATA's attention masking helps the model focus on executing the intended sub-tasks, which **not only improves instruction-following** **but may also lead to better, more coherent generation**.
> We attribute this safe attention steering to our **(1) Precise Masking**: restricting attention masking to subtask spans while preserving global context; and **(2) Targeted Steering**: steering only a small, pre-selected subset of attention heads relevant to structural reasoning, maintaining the integrity of the majority of heads responsible for core language capabilities.
>
> We will incorporate this analysis into our revision. Thank you for inspiring us to strengthen this critical aspect of our evaluation.
>
> > [1] Stolfo et al., “Improving Instruction-Following in Language Models through Activation Steering,” ICLR 2025.
>
> ---

---

> > ### Comment · Reviewer_qkLE · 2025-11-27
> > **Response to Author Comments**
> >
> > Thank you for the response to my questions. The authors have addressed most of my concerns. Since, my original rating is already positive I am maintaining my score.

---

### Meta-Review · Area_Chair_NmRQ · 2025-12-29

**Summary:**

This paper proposes Attend to the Active (ATA), an inference-time, structure-aware dynamic attention mechanism for improving compositional instruction following in large language models. The authors summarized three common compositional structures (chaining, branching, parallel), and pinpointed the key insight that **only one sub-task should be active at any generation step, and attention to inactive sub-tasks induces harmful interference**. ATA identifies the active sub-task online and applies targeted attention masking and steering to suppress irrelevant sub-tasks, without parameter updates.

Across reviewers, the paper is viewed as well motivated, clearly written, technically sound, and empirically strong. Reviewers particularly praised the lightweight inference-time design, strong ablations, robustness analyses, and breadth of evaluation across models, structures, and baselines. Initial concerns include
- realism of the parallel instruction benchmark,
- preservation of general language ability,
- clarity of active sub-task identification and control mechanisms,
- missing evaluations on general instruction datasets,
- overhead and structure-identification reliability.

The authors provided thorough and technically convincing rebuttals, adding new benchmarks, analyses, and experiments that directly addressed these concerns. **In the existing discussion, reviewers either maintained or increased their scores, and no major technical objections remain.**

**Reviewer Concerns:**

Concerns that have been addressed satisfactorily:
- In response to parallel benchmark realism by Reviewer qkLE, the authors introduced a new, more challenging Parallel-Challenge dataset with shared constraints and validated ATA’s effectiveness.
- In response to general capability preservation by Reviewer qkLE: additional evaluations following prior work showed that ATA does not degrade (and in some cases improves) general response quality.
- In response to applicability on "real" tasks by Reviewer TRL7: New results on IFEval, GSM8K, and HumanEval demonstrate ATA’s safety on simple instructions and effectiveness when combined with planning/decomposition methods.
- In response to active sub-task identification clarity by Reviewer grgz: detailed explanations of score computation, entropy thresholding, and the distinction between score-based identification and active control resolved earlier confusion.
- In response to concerns on global structure adherence raised by Reviewer grgz: the authors clarified the active control mechanism, explaining how ATA enforces safe sequential progression for chaining/parallel tasks and consistent execution for branching, supported by ablations.
- In response to concerns on branching conditions and nested structures raised by Reviewer grgz: the authors explained why conditions are treated as shared context, detailed recursive structure identification for nested tasks, and committed to including prompts and explanations in the revision.
- In response to applicability and robustness concerns raised by Reviewer Hrq3: new results on IFEval, GSM8K, and HumanEval show ATA’s safety on non-compositional instructions and effectiveness when combined with planning/decomposition methods.
- In response to efficiency and practicality concerns raised by Reviewer Hrq3: the authors provided runtime analysis demonstrating minimal inference overhead and amortizable structure-identification cost, and committed to fixing minor presentation issues.

**Reviewer Scores:**

- Reviewer qkLE: Maintained positive score (8) after rebuttal.
- Reviewer TRL7: Positive score (8), concerns fully addressed.
- Reviewer grgz: Indicated that he/she would increase his/her previous score (4) after clarification; now supports acceptance.
- Reviewer Hrq3: Marginally positive (6); concerns addressed with added evaluations.

Overall reviewer consensus converged to clear acceptance.

---

### Decision · Program_Chairs · 2026-01-26

Accept (Poster)